# MYC Ran Up the Clock: The Complex Interplay between MYC and the Molecular Circadian Clock in Cancer

**DOI:** 10.3390/ijms22147761

**Published:** 2021-07-20

**Authors:** Jamison B. Burchett, Amelia M. Knudsen-Clark, Brian J. Altman

**Affiliations:** 1Department of Biomedical Genetics, University of Rochester Medical Center, Rochester, NY 14642, USA; jamison_burchett@urmc.rochester.edu; 2Department of Microbiology and Immunology, University of Rochester Medical Center, Rochester, NY 14642, USA; Amelia_clark@urmc.rochester.edu; 3Wilmot Cancer Institute, University of Rochester Medical Center, Rochester, NY 14642, USA

**Keywords:** MYC, cancer, circadian rhythm, molecular clock, transcription factor, chromatin, pause release, tumor immunology, computational biology, cancer metabolism

## Abstract

The MYC oncoprotein and its family members N-MYC and L-MYC are known to drive a wide variety of human cancers. Emerging evidence suggests that MYC has a bi-directional relationship with the molecular clock in cancer. The molecular clock is responsible for circadian (~24 h) rhythms in most eukaryotic cells and organisms, as a mechanism to adapt to light/dark cycles. Disruption of human circadian rhythms, such as through shift work, may serve as a risk factor for cancer, but connections with oncogenic drivers such as MYC were previously not well understood. In this review, we examine recent evidence that MYC in cancer cells can disrupt the molecular clock; and conversely, that molecular clock disruption in cancer can deregulate and elevate MYC. Since MYC and the molecular clock control many of the same processes, we then consider competition between MYC and the molecular clock in several select aspects of tumor biology, including chromatin state, global transcriptional profile, metabolic rewiring, and immune infiltrate in the tumor. Finally, we discuss how the molecular clock can be monitored or diagnosed in human tumors, and how MYC inhibition could potentially restore molecular clock function. Further study of the relationship between the molecular clock and MYC in cancer may reveal previously unsuspected vulnerabilities which could lead to new treatment strategies.

## 1. Introduction to MYC and the Molecular Circadian Clock in Normal Cellular Biology

The MYC family of proto-oncogenes is a group of protein-coding transcription factors whose effects extend to various biological processes, including cell cycle progression and metabolic control. C-MYC (*MYC*), N-MYC (*MYCN)*, and L-MYC (*MYCL)* are members of the basic helix-loop-helix (bHLH) family of transcription factors [1]. These three proteins have slightly different transcriptional programs and tissue specificities, but henceforth we will refer to all three as ‘MYC’ in this review, except where otherwise specified. Canonically, MYC and its main binding partner MAX form a heterodimer which occupies E-box (CANNTG) regions of promoters and enhancers to initiate and support transcription of target genes. When MYC is amplified, as is the case in many cancers, these target genes can number in the thousands [1]. However, MYC’s role in transcription extends well beyond simple promoter binding; MYC also mediates chromatin reorganization and RNA polymerase II pause release [2]. Additionally, MYC can repress expression, either through binding and antagonizing the transcription factor MIZ1, or through other, perhaps indirect, means [2].

In actively dividing cells, MYC acts as a stimulant for cell cycle progression through multiple mechanisms, including the induction of cyclins and cyclin-dependent kinases (CDKs), downregulation of their inhibitors p21^CIP1/WAF1^ and p15^INK4B^, and promotion of the degradation of p27^KIP1^, all of which aid in cell cycle progression [3]. This ability of MYC to promote cell cycle progression is often hijacked by cancer cells to overcome quiescence and maintain active cell division [3,4,5]. In fact, at least 28% of human cancers have amplified or translocated *MYC*, *MYCN*, or *MYCL*; in many more, MYC is upregulated downstream of other oncogenic insults [4]. Given MYC’s important role in cell cycle and other processes such as metabolism, tight regulation of MYC expression in healthy cells is necessary. In this review, we discuss the molecular circadian clock as a regulator of MYC expression under normal homeostasis, how MYC itself can influence circadian rhythmicity of individual cells, and how this bidirectional regulation is potentially disrupted in cancer.

Three other members of the bHLH family—CLOCK, NPAS2, and BMAL1 (*ARNTL*)—control the molecular clock in cells, which leads to circadian rhythms in cellular and organismal activity [6]. Circadian rhythms are 24 h cycles which synchronize behavior and activity to the day / night cycle and controls ~24 h behavior in most organisms [7]. In mammals, bright light (including daylight and artificial light) is sensed by melanopsin in the eye, which signals to the suprachiasmatic nucleus of the hypothalamus, termed the ‘central clock’ [7]. This central clock directly influences certain processes such as wakefulness and alertness, but also coordinates the activity of individual ‘peripheral clocks’ present in every cell in the body, through control of synchronizing stimuli such as autonomic nerve impulses and rhythmic release of corticosteroids [8]. These peripheral clocks control oscillatory processes throughout the body, including circadian control of metabolism both on a whole-body and single-cell level [9,10]. Perturbation of the molecular clock can thus lead to negative metabolic consequences. As an example, multiple groups have shown that deletion of BMAL1 in mice, either in a whole body- or pancreas-specific manner, can lead to loss of insulin oscillation and subsequent weight gain and diabetes [11,12]. Disruption of circadian rhythms is also applicable to human physiology, as chronic jet lag or late-night shift work, both disruptions of natural human circadian patterns, weaken or ablate the body’s circadian rhythm and potentially have metabolic impacts. Chronic circadian activity disruption can lead to altered glucose and insulin sensitivity, weight gain, and other metabolic diseases [13].

What is referred to as the ‘molecular clock’ is a regulatory mechanism consisting of a group of cooperative proteins with oscillatory expression that follows a circadian rhythm and has influence on several different biological functions (Figure 1A). The molecular clock is controlled by a complex of the transcription factors CLOCK (or its paralogue NPAS2) and BMAL1, which, when duplexed, occupies E-box promoter regions [7]. CLOCK-BMAL1 controls expression of select output genes in a 24 h rhythm, though the identity of these oscillatory genes varies quite widely in a tissue-specific fashion [14,15]. While a repertoire of molecular clock genes likely exist in very large megadalton-scale complexes [16], they have been classically understood to regulate each other’s activity and localization through a series of feedback loops. CLOCK and BMAL1 upregulate the transcription of genes encoding the effector proteins PER and CRY, which then feed back to negatively regulate CLOCK and BMAL1 activity. This results in the CLOCK-BMAL1 complex losing its transcription factor potential and creates a sustaining loop [7]. Acetylation and nuclear–cytoplasmic shuttling of the PER proteins also play key roles in their inhibitory potential [17]. A secondary loop that is critical for molecular clock regulation in many tissues involves the REV-ERB (REV-ERBα and REV-ERBβ, or *NR1D1* and *NR1D2*) and ROR (RORα, RORβ, and RORγ) proteins, both of which are nuclear hormone receptors that regulate BMAL1 [18]. RORs, or retinoic acid receptor-related orphan receptors, act as transcriptional activators when bound to their response element (RRE, ROR-response element) while REV-ERBs act as transcriptional repressors when binding to the same element [7]. The REV-ERB proteins, whose ligand is heme [19], have been heavily studied, and deletion of both paralogues ablates the molecular clock across several tissues [20,21]. CLOCK -BMAL1 and MYC-MAX both have very similar heterodimer structures and bind to largely identical promoter sequences, setting up the possibility that these two transcription factor heterodimers have overlapping functions and may compete in cancer cells.

## 2. Bi-Directional Relationship between MYC and the Molecular Clock in Cancer

### 2.1. MYC’s Influence on the Molecular Clock in Somatic Cells and Cancer

The MYC family of proteins is upregulated in a large number of human cancers, but the manner of this upregulation, and whether it is direct or indirect, varies. In many cancers, the *MYC* locus is perturbed to allow for unrestricted expression. Chromosomal translocations place *MYC* under the control of novel promoter and regulatory elements. These translocation events are primary drivers in B cell cancers such as Burkitt’s lymphoma (where they occur with up to 80% frequency), diffuse large B-cell lymphoma, and multiple myeloma (where they occur in each with up to 15% frequency) [23,24,25]. In the case of Burkitt’s lymphoma, a typical translocation event places *MYC* under the control of the *IGH* promoter, leading to both loss of normal transcriptional control of *MYC*, and massive overexpression [25]. In other cancers including solid tumors, *MYC* and its family members are instead subject to focal amplification, where the *MYC* locus appears dozens to hundreds of times throughout the genome in inappropriate locations. In a recent study that queried genomic cancer patient data from The Cancer Genome Atlas, *MYC* or one of its paralogues was found to be amplified in 28% of all cancers. However, this number is much higher in specific cancers; for instance, in ovarian cancer, one of the three *MYC* paralogues is amplified in nearly 100% of cases [4]. Similarly, amplification of enhancer or super-enhancer regions may also deregulate MYC expression [26]. In such cases of translocation or amplification of coding or enhancer regions, MYC likely loses endogenous control of its expression [4,26]. Throughout this review, when we refer to oncogenic MYC, we will generally be referring to these instances where *MYC* is amplified or rearranged in the genome, or where cell line or mouse models simulate this amplification.

In contrast, MYC can be stabilized or upregulated downstream of other oncogenic mutations. To illustrate a few examples, *MYC* can be upregulated by the BCR-ABL fusion protein in chronic myelogenous leukemia, mutant Notch in T cell acute lymphoblastic leukemia, PTEN loss in breast cancer, and EGFR mutation in non-small-cell lung cancer [27,28,29,30,31,32,33]. Unlike in the case of genomic amplification or translocation, the promoters and regulatory elements that govern *MYC* regulation and expression are intact when it is upregulated downstream of other mutations. The manner of MYC upregulation in cancer, whether it be genomically altered, or upregulated downstream of other oncogenic mutations, is important to consider in the context of this review. *MYC* and the molecular clock extensively cross-talk, and the outcome of this relationship in cancer may depend on how MYC is deregulated.

Emerging data suggest that MYC and the molecular clock have a fascinating bidirectional relationship which is likely perturbed in many cancers. Several recent studies across multiple organisms and model systems have addressed the interplay of oncogenic MYC and circadian rhythm, and these studies have overwhelmingly suggested that deregulated MYC disrupts or fully ablates oscillation of the molecular clock. These mechanisms are summarized in Figure 1B. Two sets of work suggest that overexpressed MYC suppresses BMAL1, leading to loss of oscillation of molecular clock components [34,35,36,37,38], albeit through slightly different mechanisms. In one mechanism, overexpressed MYC directly upregulated REV-ERBα, REV-ERBβ, and other molecular clock components by binding to the same E-box site normally occupied by CLOCK-BMAL1. The resulting REV-ERB upregulation led to BMAL1 suppression and loss of molecular clock function [34,35]. A second mechanism involves MYC antagonism of the transcription factor MIZ1. MIZ1 transcriptionally upregulated BMAL1 (*ARNTL)*, *CLOCK*, and the CLOCK paralogue *NPAS2* [36,37,38]. When MYC bound to MIZ1 and antagonized its transcriptional activity, molecular clock function was ablated [36,37,38]. These two groups of studies showed that MYC suppressed BMAL1 across a wide range of cancers, including osteosarcoma, neuroblastoma, hepatocellular carcinoma, Burkitt’s lymphoma, and T cell acute lymphoblastic leukemia [34,35,36,37,38]. Several other putative mechanisms exist whereby MYC disrupts the molecular clock. In some cancers, MYC occupied molecular clock promoters and directly repressed rather than transactivated molecular clock gene expression [39]. Intriguingly, in embryonic stem cells, deregulated MYC did not directly modulate expression of molecular clock genes, but rather promoted disrupted PER nuclear–cytoplasmic shuttling, again resulting in ablation of circadian oscillations in gene expression [40]. The ability of MYC to disrupt the clock is not limited to vertebrates; a recent study demonstrated that overexpressed dMyc in *Drosophila* bound to molecular clock gene promoters, and elevated dMYC ablated behavioral and metabolic rhythms [41]. It is again important to note that in all the above studies, genomically re-arranged or amplified MYC rather than upregulated endogenous MYC was being recapitulated in each of the model systems. The contrasting roles of genomically amplified MYC and overexpressed endogenous MYC on the molecular clock are discussed in more detail in Section 2.3, below.

The role of endogenous MYC in regulating the molecular clock is far less clear, and largely remains to be uncovered. MYC is upregulated when cells are stimulated with mitogens to enter the cell cycle [1], and one study suggested that mitogen-stimulated MYC may bind to BMAL1 [39], though the function of this novel heterodimer remained undetermined. Separately, another recent study set out to determine the role of MYC-MAX in control of the molecular clock, and unexpectedly found that MAX exerts a strong MYC-independent role in repressing molecular clock genes as well as CLOCK-BMAL1-regulated output genes [42]. MAX forms heterodimers with other MYC-family proteins besides MYC, N-MYC, and L-MYC, including the repressive MAD family [1], and the observed repressive effect of MAX was potentially due to a MAD-MAX complex. These studies provide interesting context for the role of endogenous MYC in circadian control, but detailed genetic and chemical inhibition of endogenous MYC in non-transformed cells will be required to more clearly understand how it interplays with the molecular clock.

### 2.2. Links between a Deregulated Circadian Clock and Cancer, and Downstream Effects on MYC Expression

A link between circadian disruption, tumorigenesis, and patient outcomes in cancer has long been suspected. Analysis of the Nurses Health Study I and II, each representing more than 70,000 volunteers, was performed observing women who worked third/night shifts, causing a them to develop a disrupted circadian rhythm. Through the study, it was noted that these women with disrupted circadian rhythms had increased incidence of breast and lung cancer compared to their counterparts [43,44,45]. In fact, the International Agency for Research on Cancer (IARC) now classifies night-shift work as a probable carcinogen [46]. Separately, in cancer patients and survivors, disrupted rhythms in activity, or serum cortisol and melatonin, are predictive of poor outcome in survivors of lung, breast, and colorectal cancer [47,48,49,50,51,52]. These findings have since been supported by a number of preclinical animal studies showing that simulated shift work or jet lag initiates and accelerates tumorigenesis in mouse and rat models of non-small-cell lung cancer, liver cancer, osteosarcoma, and melanoma [53,54,55,56,57,58,59]. Similarly, genetic mutation or ablation of clock components and subsequent deregulation of the molecular clock also accelerated tumorigenesis [56,57,60,61]. These findings have direct clinical implications, as restoring rhythmicity through glucocorticoid stimulation may slow tumor growth or aid in treatment with anticancer therapeutics [58,62,63]. An interesting consequence of these studies was the unexpected revelation of a deep and complicated relationship between the molecular clock and *MYC*.

The molecular circadian clock and MYC have a bi-directional relationship: while oncogenic MYC can disrupt the molecular clock, it appears that the molecular clock itself can regulate *MYC* expression. What we know of molecular clock regulation of *MYC* comes mostly from tumorigenesis studies; the role of the molecular clock in regulating endogenous *MYC* in normal and healthy cells remains to be determined. Nonetheless, the consensus amongst multiple studies is that genetic disruption of the molecular clock tends to de-repress and upregulate *MYC*. A study in mouse liver revealed that *Myc* exhibits circadian oscillation, and was upregulated up to 25-fold (but remained oscillatory) when a dominant-negative PER2 mutant was expressed [60]. Similarly, *Per2* mutation and *Arntl* (BMAL1) deletion in lung tumors both resulted in upregulated *Myc* [57]. These results were recapitulated in chronic jet lag models of liver cancer and osteosarcoma, where exposure to chronic jet lag or desynchrony resulted in elevated *Myc* expression and activity [56,58]. However, the manner in which the molecular clock is disrupted dictates its regulation of MYC. In a recent study of molecular clock knockout mutants from mouse spleen, BMAL1 deletion led to MYC upregulation [64], as seen in other studies. In contrast, while individual deletion of *Cry1* or *Cry2* had little effect on *Myc* mRNA expression or MYC protein levels, *Cry1*/*Cry2* double knockout (which also ablates the molecular clock) instead led to MYC repression. The authors attributed this to CLOCK-BMAL1 being locked in an active state without the negative feedback of the PER-CRY complex, which indirectly led to *Myc* suppression [64].

These studies suggest that CLOCK-BMAL1 may repress *MYC* expression, but the molecular clock exerts control over MYC in other ways as well. Studies in synchronized neuroblastoma cell lines showed that histone modifications control oscillatory *MYC* expression, and that MYC protein oscillation was modulated by circadian control of protein stability [39,65]. It was later discovered that ubiquitination and ultimate turnover of MYC protein is directly regulated by CRY2, which participates in an SCF E3-ubiquitin ligase complex targeting MYC. Deletion of *Cry2* led to elevated MYC protein in transformed mouse fibroblasts, human lung cancer and colon cancer cell lines, and a mouse model of lymphoma [61]. Importantly, despite being upregulated by *Cry2* deletion, MYC still retained circadian oscillation in mouse fibroblasts. CRY2′s role in regulating MYC protein stability may be cell type and context specific, as *Cry2* deletion in mouse spleen did not affect MYC protein levels, despite components of the SCF E3-ubiquitin ligase complex being present in this tissue [64]. More studies are clearly required to delineate the specific conditions under which CRY2 regulates MYC protein stability.

### 2.3. How and When Does MYC Disrupt the Molecular Clock? Context Matters

A unifying theme emerges from the above studies: even though MYC was elevated due to behavioral or genetic clock disruption, when oscillation of the molecular clock was intact, MYC continued to oscillate as well. This can be seen most clearly in PER2-mutant overexpression in the liver, and *Cry2* deletion in mouse fibroblasts [60,61]. In both these cases, MYC was upregulated at least 25-fold as compared to wild-type cells; however, because neither *Per2* mutation nor *Cry2* deletion ablated the molecular clock, MYC continued to oscillate [60,61]. Interestingly, the converse is also true: in these models, even though MYC is greatly elevated, it still oscillates and is thus is unable to ablate molecular clock oscillation. This is in stark contrast to models of MYC amplification or translocation, where static (non-oscillatory) overexpression of MYC results in ablation of molecular clock oscillation across multiple models of transformed and non-transformed cells [35,37,39,40]. This raises the idea that when MYC is upregulated may be just as important as how much it is upregulated. If MYC remains circadian downstream of other oncogenic insults, it may have a far subtler role in modulation of the molecular clock. Thus, when diagnosing in human cancers whether MYC disrupts the clock, mutation status of the *MYC* oncogenes themselves may be critical in determining its role in circadian oscillation.

A parallel field of studies suggests that in some cancers distinct from the types discussed above, the molecular clock may promote, rather than restrain tumorigenesis. This was demonstrated chiefly in cancers that have a distinct stem cell component, such as skin cancer, acute myelogenous leukemia, and glioblastoma [66,67,68]. These stem and cancer cells retain robust rhythms, and deletion of core clock components such as BMAL1 restrained or even reversed tumorigenesis, particularly in stem-like cells. Interestingly, suppression of either CLOCK or BMAL1 in glioblastoma stem cells (GSCs) led to a decrease rather than an increase in MYC levels [68], unlike what was observed in liver, lung, and spleen [56,60,64]. As *MYC* is known to be a critical factor for maintenance of stem cells in general and GSCs in particular [69,70], it is interesting to speculate that MYC and CLOCK-BMAL1 cooperate in stem-like cancer cells to maintain a state of stemness [68]. Indeed, the authors noted that BMAL1 bound to the promoter of *MYC* in GSCs [68], but did not show whether *MYC* oscillated, which may be an intriguing topic of future study.

## 3. Control of Chromatin State and Pause Release by MYC and the Molecular Clock

### 3.1. Role of MYC in Chromatin and Global Pause Release

bHLH transcription factors such as MYC and CLOCK-BMAL1 directly influence chromatin state and pause release. The interaction between MYC and CLOCK-BMAL1 in controlling chromatin and pause release in healthy cells is unknown; however, evidence suggests that their mutual regulation of these processes is out of balance in cancer. MYC may play a role in global chromatin remodeling, such as during B cell activation [71,72]; however, the literature suggests that MYC specificity, especially in cancer cells, relies on the pre-existing availability of open chromatin and chromatin modifying enzymes permissive to transcription. Large-scale comparison of chromatin immunoprecipitation and RNA sequencing data revealed that MYC preferentially bound to certain E-box regions over others, such as those that have high levels of H3K4me3, a histone modification associated with more open chromatin [73,74]. While MYC has many binding partners that modulate its function and output (reviewed recently in [75]), one factor particularly important in determining MYC binding is WDR5, which is involved in remodeling chromatin into a more active state [76]. Several studies have identified that WDR5 presence on chromatin is critical in permitting MYC to bind to DNA, and for MYC-driven tumorigenesis [74,77] (Figure 2). WDR5 also plays a role in PER-mediated repression of CLOCK-BMAL1 [78], and thus may serve as a link between MYC and CLOCK-BMAL1 binding to common promoter sites.

Oncogenic MYC has a role not just in transcriptional initiation, but also in RNA polymerase II (Pol II) pause release. Pause release is the process whereby Pol II ‘pauses’ shortly after initiating transcription, and is later released by external factors to complete transcriptional elongation of the target gene [79]. Recent literature strongly suggests that oncogenic MYC promotes global Pol II pause release. Furthermore, it has been proposed that this global pause release and amplification of transcription is responsible for the majority of MYC activity in cancer and other settings of MYC hyperactivity [80,81,82,83]. Indeed, pause release and transcriptional elongation have been considered as possible vulnerabilities in MYC-driven cancer cells that could be exploited as drug targets [84,85]. However, further studies revealed that while this view of MYC as a universal amplifier of transcription was valid in many settings, it was also incomplete. This was due to several observations. First, MYC is known not only to upregulate certain transcripts, but also to repress others [2,86,87]. In some cases, this is through antagonism of the transcription factor MIZ1, and in other cases, the cause for MYC-driven transcriptional repression is not well understood. In fact, this association with MIZ1 to repress transcription (or, in the case of N-MYC, with BRCA1) may be required to buffer extreme global transcriptional elongation which would otherwise lead to stress and DNA damage [86,88]. Second, MYC invasion of promoters and enhancers is selective, and depends not only on chromatin state and the presence of factors such as WDR5, but also on site specificity and total MYC levels. This is due to the promiscuity of MYC when it is present at higher levels, which results in binding to sequences for which it has a lower affinity [2,74,87,89]. The current consensus is that MYC amplifies transcription but also executes a specific program of metabolism and biomass generation that requires both the activation and repression of certain genes [2].

### 3.2. Molecular Clock Control of Chromatin State and Pause Release

Histone modifications (and by extension, chromatin state) have long been known to oscillate [90], and more recently, it was shown that several components of the molecular clock can directly influence chromatin state and nuclear organization. CLOCK and BMAL1 themselves function as pioneer-like transcription factors and can rhythmically direct repositioning of nucleosomes, while REV-ERBα contributes to rhythmic reorganization of chromatin looping and topologically-associated domains [91,92]. Rhythmic promoter–enhancer interactions also contribute to rhythmicity of clock output genes [93]. Although the molecular clock machinery contributes to the process of rhythmic chromatin reorganization, evidence suggests that this alone is not sufficient to independently upregulate gene expression [94,95]. Instead, the molecular clock interacts with tissue-specific transcription factors and chromatin states in a cooperative fashion to effect change in chromatin organization and gene expression [94,95].

The relationship between the molecular clock and polymerase II pause release is somewhat less clear, and still a highly active area of research. It has long been appreciated that CLOCK and BMAL1 rhythmically recruit Pol II and associated factors to promoters; this was thought be a principal driver of rhythmic transcription [90,96]. However, recent studies suggest that pause release may also play a major role in rhythmic transcription. Computational analysis suggests that in mammals, especially strong circadian promoters with low nucleosome content and directly bound by circadian clock machinery are likely heavily influenced in their rhythmicity by pause release. This was confirmed in mouse liver and in fibroblasts, where circadian pause release controlled many rhythmic transcripts, and that BMAL1 itself participated directly in the regulation of the pausing machinery [97,98].

### 3.3. How Do MYC and the Molecular Clock Interact to Regulate Global Transcriptional Output?

The ability of oncogenic MYC to shape global chromatin organization and transcriptional elongation may also impact the ability of the molecular clock to govern these same processes, which could lead to competition between these two sets of transcriptional machinery. Given that oncogenic MYC is reliant on chromatin state for specificity, circadian regulation of chromatin state may impact which genes MYC can upregulate, and when. In cancer cells where the molecular clock is retained, oscillation of chromatin state, availability of chromatin-binding proteins like WDR5, and higher-level chromatin organization may still occur [78,90]. We speculate that oscillating chromatin state and presence of CLOCK-BMAL1 on promoter sequences will limit which sequences are available for MYC to regulate. Similarly, to the extent that CLOCK-BMAL1 participate in pause release, their rhythmic control of pause release may also compete with MYC’s general pause-releasing capabilities.

In contrast, in settings where oncogenic MYC ablates the molecular clock, the entire chromatin landscape may change, allowing for MYC to enact an even more transformative program in tumor cells. MYC’s ability to statically (irrespective of time-of-day) upregulate proteins like REV-ERBα and suppress BMAL1 may suppress or even eliminate the ability of the molecular clock to rhythmically regulate chromatin accessibility and higher-level organization. This, along with MYC’s ability to promote global pause release, may unleash a novel transcriptional program quite unlike that in cells with a functional molecular clock. These conjectures suggest that simply quantifying MYC expression in a given tumor may not be enough to properly diagnose MYC’s potential impact on the molecular clock in human cancer. Assessing potential *MYC* genomic amplification, and using computational approaches to assess the functionality of the molecular clock in tumor cells (discussed in more detail below) will be required to fully appreciate the role of MYC in modulating circadian rhythms in human cancer cells.

## 4. MYC and CLOCK Clash for Control of Metabolism

### 4.1. Brief Overview of MYC-Mediated Metabolic Rewiring

Cell-autonomous control of metabolism is a key node where MYC and the molecular clock may compete for control of cell physiology. MYC is well known to rewire metabolism to promote a constantly proliferative state. The role of MYC in rewiring metabolism has been previously reviewed [5]; but briefly, MYC is known to drive glucose uptake, glycolysis, and lactate secretion to fuel biosynthetic pathways, while also increasing uptake of glutamine and other amino acids such as arginine [99] to fuel anaplerosis. More recently, it has been shown in multiple studies that MYC drives fatty acid biosynthesis in multiple cancers, in some cases in cooperation with the MYC-family member MONDOA [100,101]. As discussed below, this dramatic metabolic rewiring uncovers potential vulnerabilities in MYC-driven cancer cells.

### 4.2. The Molecular Clock Controls Cell Autonomous Metabolic Oscillation and Metabolic Gene Expression Programs

In contrast to the well-known role of MYC in driving proliferative metabolism in cells, the role of the molecular clock in cell-autonomous metabolism has only been addressed recently. It has been known for decades that the circadian clock strongly controls whole body metabolism, particularly pancreatic endocrine and liver functions [9]; however, it was unclear if the molecular clock controlled individual cell (cell-autonomous) metabolic oscillations. Red blood cells and fibroblasts were shown to have cell-intrinsic circadian oscillations in redox state, which were more recently found to be dependent on rhythmic glucose metabolism [102,103]. These findings agree with recent work showing that intracellular glucose, nucleotide metabolism, amino acid metabolism, and the 1-carbon pathway all oscillate in a cell-autonomous circadian fashion across cell lines, cultured mouse hepatocytes, and mouse liver [10,35]. More recent work in the field has focused on how circadian rhythms regulate higher-order metabolite biosynthesis and pathways that influence cellular metabolism. Circadian oscillations in ATP production and oxidative phosphorylation were found to arise from rhythmicity in mitochondrial fission and fusion [104]. Similarly, lipid content and dynamics in the mitochondria and nucleus were found to be controlled by the molecular clock [105]. While CLOCK and BMAL1 likely directly control the oscillation of certain metabolic gene transcripts, they also cooperate with master regulators of metabolism to influence cellular metabolic oscillations. The molecular clock was found to control the redox-sensitive NRF2 transcription factor, which influences downstream redox state and nutrient uptake [106], and the clock was also shown by multiple groups to cooperate with the hypoxia-sensitive transcription factor HIF1α [107,108,109], which may influence downstream oscillation of glucose metabolism and other pathways that respond to low oxygen stress. The burgeoning appreciation of cell-autonomous circadian metabolism has been shown to have impacts reaching as far as immune cell function; macrophages, which are phagocytic white blood cells of the innate immune system, were recently shown to possess cell-intrinsic oscillations in mitochondria morphology, TCA cycle intermediates, and NRF2 activity, influencing downstream phagocytic and inflammatory responses [106,110].

### 4.3. How Does MYC-Driven Metabolic Rewiring Interact with Circadian Oscillation of Cell Metabolism?

If amplified MYC disrupts molecular clock oscillation, it may also disrupt cell-autonomous metabolic oscillations; indeed, it was demonstrated that intracellular glucose oscillations are ablated by MYC overexpression in cancer cells [35]. MYC-driven tumors are well known to be more sensitive to nutrient deprivation and metabolic inhibitors as compared to other tumors [5]. Thus, the status of the molecular clock in these cells may inform how these nutrient deprivation strategies are best used. Many types MYC-amplified such as glioma, Burkitt’s lymphoma, and neuroblastoma are known to rely on increased glutamine uptake and glutaminolysis, and rapidly die by apoptosis upon glutamine starvation [111,112,113]. MYC was also shown to drive glutamine uptake in animal models of hepatocellular carcinoma and clear cell renal cell carcinoma [114,115]. Glutamine is not the only key amino acid for MYC-driven cancers: it was recently observed in MYC-induced small-cell lung cancer that these tumors depend on arginine uptake and metabolism [99]. Depletion of arginine through chemical means had an effect similar to that of glutamine withdrawal, suppressing in vivo tumor growth [99].

Metabolic pathways, such as glutamine and arginine uptake and catabolism, likely oscillate in non-transformed cells as well as some cancer cells with an intact clock [10]. These lead to several interesting questions. First, it remains to be seen whether an intact clock, such as in cells where MYC is elevated but still oscillatory, confers a time of day-dependent vulnerability to nutrient depletion or metabolic therapy. Second, in cancer cells where amplified MYC has ablated the clock, it is not well understood if clock ablation is directly tied to metabolic rewiring (perhaps through a change in chromatin state, as discussed above), or instead if these two phenomena occur simultaneously but independently. A recent study addressed this question in neuroblastoma cells where amplified N-MYC suppresses BMAL1 expression [38]. In this context, pharmacologically restoring BMAL1 expression led to suppression of N-MYC-driven lipogenesis and diminished tumor growth [38], suggesting that N-MYC and CLOCK-BMAL1 may drive opposing metabolic programs in these cancer cells. Regardless of how MYC and CLOCK-BMAL1 are connected, it remains to be determined how molecular clock disruption correlates with vulnerability to nutrient depletion in cancer. These questions also highlight the potential importance of diagnosing clock function in MYC-overexpressing cancer cells, discussed in more detail below.

## 5. Does Time Stop When MYC Alters the Tumor Immune Microenvironment?

### 5.1. MYC Modulation of the Tumor Immune Microenvironment

Tumors must escape detection by the immune system as foreign bodies in order to successfully develop and progress [116]. Immune evasion is necessary not only for tumor growth, but also for persistence through resistance to treatment. This is illustrated by studies showing that tumor cell killing following treatment with radiation or chemotherapy is maximized only when there is a robust antitumor immune response [117,118,119]. MYC has long been known to modulate the immune microenvironment of tumors, but how its disruption of the molecular clock plays into this is not known. The first evidence that oncogenic MYC may suppress the antitumor immune response was the finding that MYC-expressing cancer cells downregulate major histocompatibility complex protein I (MHC I), which is critical for cytotoxic CD8+ T cells to recognize and specifically kill tumor cells [120,121,122,123]. Since that time, it has been increasingly apparent that MYC has the potential to shape the immune infiltrate of the tumor microenvironment in different ways. Amplified MYC was recently shown across several tumor models to directly upregulate CD47 and PD-L1 in cancer cells, which served to suppress recruitment and promote exhaustion of T cells [124]. Another group showed that MYC suppressed CCL5, which may normally mediate recruitment of T cells to tumors [124,125]. Downregulation or indirect activation of MYC or N-MYC led to a repression of PD-L1 in the tumor and an influx of antitumor CD4+ T helper cells that correlated with tumor regression [126,127]. Interestingly, MYC may drive these changes even in cancers where it is not mutated but merely upregulated downstream of other oncogenic insults [128]. MYC not only suppresses the recruitment of antitumor immune cells, but may also selectively recruit immune cells that can be polarized toward a pro-tumorigenic phenotype at the site of the tumor. Amplified MYC is known to drive recruitment of mast cells that participate in angiogenesis [129], and it was more recently shown in a non-small cell lung cancer model that amplified MYC, in cooperation with mutated oncogenic K-RAS, drove depletion of T cells and natural killer (NK) cells from the tumor which corresponded with intra-tumoral accumulation of pro-tumirogenic CD206+ macrophages with elevated expression of PD-L1 [123]. This accumulation of pro-tumorigenic macrophages aided in angiogenesis, and likely also promoted the depletion of T cells from the tumor via induction of anergy, as has been shown to occur in other tumor models [123]. A summary of these mechanisms is presented in Figure 3A,B. In these ways, oncogenic MYC can shape the tumor immune microenvironment to support influx and polarization of pro-tumorigenic immune cells and promote exclusion or exhaustion of antitumor cells.

### 5.2. Connections between MYC and Potential Molecular Clock Disruption in the Tumor Immune Microenvironment

While it is clear that oncogenic MYC can affect the molecular clock of tumor cells, it is less clear how MYC can influence the clock of tumor-infiltrating immune cells. A full discussion of the extent of circadian control of immune responses, and the implications of circadian disruption in immune cells is beyond the scope of this paper, but has been covered in several comprehensive reviews [130,131]. In short, current studies suggest that genetic or behavioral clock disruption in immune cells leads to an elevated basal activation state in innate immune cells, resulting in a heightened response to pathogen challenge which can be either beneficial or detrimental to the host, depending on the infection model [132,133,134,135,136]. In fact, two recent and complementary studies showed that either chronic jet lag or genetic ablation of BMAL1 in myeloid cells led to an increase in pro-tumorigenic tumor-associated macrophages and faster tumor growth [59,137,138]. How might oncogenic MYC affect the clock of neighboring immune cells? One fairly straightforward mechanism is through the MYC-mediated upregulation of the immunomodulatory proteins PD-L1 and CD47, as discussed above. If MYC suppresses BMAL1 expression, [34,35,37], it is reasonable to speculate that MYC-driven CD47 and PD-L1 on tumor cells are locked into static, non-oscillatory expression state. It remains to be uncovered how exposure to elevated and non-oscillatory CD47 and PD-L1 may affect the molecular clock and downstream functions of tumor-infiltrating immune cells.

A more subtle and indirect mechanism whereby MYC may disrupt the clock of neighboring immune cells could be due to metabolic competition and metabolic waste in the tumor (reviewed in [139]) (Figure 3C). Because tumor cells have a deregulated metabolism rewired for biosynthesis and proliferation, they uptake nutrients at a higher rate than the stromal cells around them, secrete waste products, and thus create a harsh environment that may feedback to negatively impact the molecular circadian clock of tumor immune cells. This is especially true for tumor cells with amplified MYC, which are known to simultaneously be highly glycolytic and net importers of amino acids [115,140]. Indeed, several groups have shown that high glycolytic rate, in some cases driven by MYC, deprives tumor-infiltrating T cells of glucose, suppressing their antitumor activity [141,142,143]. Separately, it was also shown that high lactate, a product of aerobic glycolysis that is excreted from cells into the microenvironment as waste and present at high levels in tumors [144], can skew macrophages towards a more pro-tumorigenic phenotype [145]. MYC can also increase amino acid uptake in tumor cells followed by subsequent local methionine depletion due to tumor cell uptake, leading to impaired intra-tumoral T cell activity [146]. It is well known that metabolic stresses found inside a tumor, such as glucose deprivation, acidosis, or hypoxia can impact or ablate the molecular clock [107,108,109,142,147], but how these metabolic stresses, often driven by MYC hyperactivity, impact the molecular clock and subsequent activity of tumor infiltrating immune cells is not known. It is likely that the circadian state of tumor-infiltrating immune cells is important to their function, as several studies have shown that cytotoxic CD8+ T cells are highly responsive to circadian stimuli, which control their peak of activity in response to stimuli [148,149]. Understanding how MYC hyperactivation affects the circadian clocks of stromal cells in the tumor will be critical in understanding how it may shape a pro-tumorigenic microenvironment.

## 6. Clinical Applications: MYC Inhibition, and Measurement of the Molecular Clock in Patients

### 6.1. Recent Strategies to Inhibit MYC in Human Cancer

In this final section, we will briefly consider novel strategies to target MYC in cancer (reviewed extensively elsewhere [150,151,152,153]). We will then turn our attention to the implications of this MYC inhibition on the status of the molecular clock. MYC and the related N-MYC and L-MYC are amplified, translocated, or mutated across a large number of human cancers, yet for many years they were considered poor targets for direct therapeutic intervention, termed “undruggable” [153]. This is for several reasons: first, MYC is not an enzyme or hormone receptor and thus does not have a classical binding pocket to target; and second, MYC has many extensive protein:protein interfaces that preclude simple inhibitor design [153]. Early demonstration of the potential efficacy of MYC inhibition came with the development of OmoMYC, an artificial MYC dominant-negative protein that effectively halts MYC-dependent transcription and leads to arrest and death in a wide variety of cancers [154,155,156,157]. While these studies began as demonstrations of the potential of MYC inhibition, today OmoMYC and related dominant negative proteins are being studied for direct clinical application [158,159]. Other approaches focus on less direct inhibition of MYC. For instance, a large class of novel anticancer therapeutics directly target bromodomain and extraterminal (BET) proteins, which act as histone readers and co-activators to increase transcription [160]. It has been long appreciated that BET inhibitors suppress *MYC* expression and destabilize MYC protein [161,162,163]. However, it continues to be unclear whether BET inhibitors exert their anticancer influences through MYC regulation or through myriad other effects they have unrelated to MYC modulation. As discussed above, MYC relies on WDR5 for binding to high affinity sites, and disruption of the MYC-WDR5 interaction may be a viable approach to treating MYC-driven cancers [77]. Finally, several groups have used small molecule inhibitors, or covalent ligation of inhibitory molecules, to directly inhibit MYC protein interactions or DNA binding [150,152,164]. These small molecules and complementary approaches have shown enormous promise in preclinical studies, but challenges in specificity in vivo bioavailability, and delivery remain. Nonetheless, given the recent flood of interest in MYC inhibition, a future where MYC-amplified cancers are targeted directly in patients seems increasingly likely.

### 6.2. Restored Rhythms after MYC Inhibition? New Strategies to Detect Molecular Clock Rhythmicity in Tumors and Other Tissues of Cancer Patients

Disruption of the molecular circadian clock is a common feature of MYC amplification across several kinds of cancer, so it stands to reason that inhibition of MYC in cancer may re-activate the molecular clock, or more precisely, allow the molecular clock of cancer cells to again respond to external entrainment signals. This raises several interesting questions: first, why would this be important, and second, how would this clock reactivation be assessed? As to the first question, whole-body circadian rhythms are known to strongly influence time-dependent drug delivery and metabolism, a field known as chronotherapeutics [165,166]. Chronotherapy, or timed delivery of chemotherapeutics for maximum efficacy and minimum toxicity, has been applied in humans with varying degrees of success [167]. Much attention has been given to assessing patient metrics such as personal activity patterns in determining optimal chronotherapeutic schedule [168,169]. In addition, understanding the circadian state of the tumor itself, including whether it has a MYC-disrupted molecular circadian clock or a more intact clock as a result of MYC inhibition, will be critical to understanding when to best deliver chemotherapy to best target the tumor while limiting patient toxicity.

Despite the importance of assessing the health and function of the molecular circadian clock in human cancers, it is quite difficult to actually conduct these measurements. Thus, with current technology, it would be challenging to determine if MYC inhibition restored circadian oscillations in a patient tumor. Ideally, to properly assess a circadian oscillation (for instance, in activity, melatonin oscillation, or mRNA oscillation), a researcher conducts serial measurements in 2–4 h increments across several days, and then tests for oscillations in these time-series data. Indeed, this technique is quite feasible with serial sampling of human blood from volunteers, and has revealed that night shift work may be a risk factor for DNA damage that can lead to cancer [170,171]. However, serial sampling of most solid tumors is impossible due to ethical and practical considerations. Therefore, for solid tumor biopsies, only single-timepoint samples are available. Several approaches have been designed to glean circadian information from these samples. One approach, “Clock Correlation Distance” (CCD), constructs a Spearman Correlation based on the known predictable relationship of twelve circadian genes, and uses an ideal correlation as a comparison point to assess the clock in human tumor data [172]. Another approach, “Cyclic Ordering by Periodic Structure” (CYCLOPS), uses machine learning to temporally order untimed health tissue and tumor samples from groups of patients, and then uses these comparisons to assess circadian rhythmicity [18]. Both CCD and CYCLOPS were highly successful at showing, for the first time, that circadian rhythms are disrupted in patient tumors and not just preclinical cell line and mouse models; however, given that both of these methods are at the population level and require large cohorts of patients to execute, in their current forms they are not a viable way to assess the state of the clock in individual patient tumor biopsies [18,172].

An alternate approach may be to assess the state of the circadian clock before and after MYC inhibition not in the tumor, but in a more easily accessed part of the body such as the blood. There is evidence from mouse models that primary tumors may distally disrupt circadian clocks throughout the body [173,174]. In human studies, patients with disrupted circadian activity, melatonin, or cortisol rhythms (all of which are distal processes to most tumors) had poorer prognosis across a wide variety of cancers [47,48,49,50,51,52]. While it is not known whether MYC-driven cancers specifically disrupt whole body circadian rhythms, these findings raise the possibility that MYC inhibition could improve the rhythmicity of cancer patients outside of their tumors. Several groups have studied the possibility of using single-timepoint patient blood samples to determine “circadian time”. While a ‘molecular timetable’ method based on blood microarray data has been available for almost two decades [175], recent advances in RNA sequencing, human subject study design, computing power, and machine learning have greatly advanced this nascent field. Three recent studies each identified a set of 15–100 genes whose expression from circulating white blood cells (leukocytes) can be used to approximately identify the time the blood sample was taken [176,177,178]. This information can then be compared to actual collection time to assess relative systemic circadian rhythm disruption. These approaches promise to allow researchers to better understand how tumors disrupt whole-body circadian rhythms and how specific treatments, such as MYC inhibition, may affect this disruption. However, limitations in these techniques preclude them from easy adoption in the clinic: two of the techniques, ZeitZeiger and Partial Least Squares Regression, while producing patient-level data, still require complete control and experimental patient cohorts to train their machine learning algorithms [176,177], precluding diagnosis of individual patients. The third technique, TimeSignature, requires two blood samples spaced approximately twelve hours apart [178], which is impractical in many clinical settings. A potential solution to these problems may come from focusing on specific populations of white blood cells rather than all leukocytes; indeed, a recent technique called BodyTime is able to accurately predict circadian time from a single blood sample provided that monocytes, a specific type of circulating white blood cell, are purified and isolated from the blood prior to analysis [179]. Overall, the field is very close to being able to deploy these techniques in the clinic to assess circadian metrics from single blood samples. This will tell us both how tumors disrupt blood rhythms, and whether therapy improves these disruptions.

## 7. Conclusions and Future Outlook

While MYC has been the subject of numerous studies investigating its multiple different interactions and effects in cancer, there is still much to be elucidated about its interplay with the molecular circadian clock. MYC disruption of the molecular clock is likely to influence metabolic reprogramming, chromatin state and accessibility, and immune infiltrate. There are still a wide range of discoveries to be made within the field, from further elucidating the role of MYC in affecting the circadian clock of cells in the tumor microenvironment to establishing the clinical value of MYC inhibition and further developing methods to assess patient circadian rhythms. Research on the MYC oncoproteins spans a variety of disciplines, and their further study is key to understanding and rationally treating MYC-driven cancers.

## Figures and Tables

**Figure 1 ijms-22-07761-f001:**
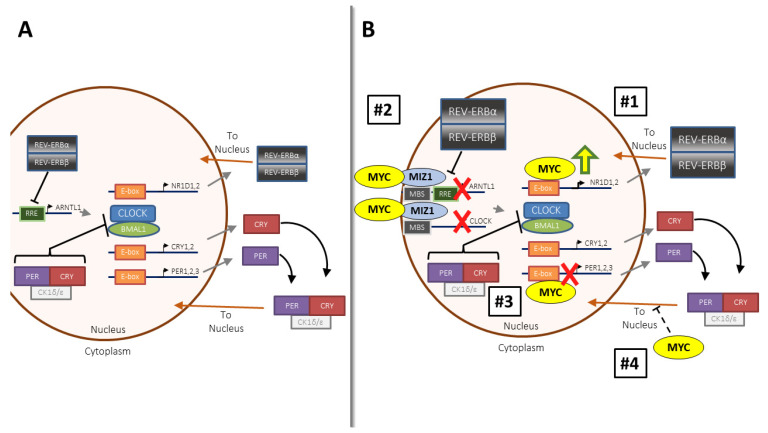
MYC disrupts the molecular circadian clock through different mechanisms. (**A**). Model and schematic of the molecular clock in mammalian cells. The two principal feedback loops governing rhythmicity are shown. Briefly, PER and CRY govern CLOCK-BMAL1 or NPAS2-BMAL1 activity, while REV-ERBα and REV-ERBβ suppress BMAL1 (*ARNTL*) transcription. The ROR (RAR-related orphan receptor) family of transcription factors positively control BMAL1 transcription (not shown). (**B**). Four mechanisms of MYC-induced disruption of the molecular clock. #1: MYC upregulates REV-ERBα and β, which suppress BMAL1 transcription. #2: MYC binds to the transcription factor MIZ1 and inhibits it, leading to suppression of CLOCK and BMAL1 transcription. #3: MYC occupies the promoters of molecular clock genes such as the *PER* genes and suppresses their transcription. #4: MYC interferes with the nuclear–cytoplasmic shuttling of the PER proteins. In each panel, grey arrows indicate expression of gene and translation to protein, orange arrows indicate translocation of protein to nucleus, black arrows indicate CRY and PER forming a complex, black suppression arrows indicate repression of target process by proteins (i.e., REV-ERBα and REV-ERBβ repress transcription of *ARNTL1*), black suppression arrows with dotted line indicate that the mechanism of MYC repression of PER translocation to the nucleus is not well defined, green and yellow up arrow indicates that MYC induces expression of gene, and red X marks indicate that expression of gene is suppressed or abolished. Abbreviations: RRE: ROR/REV-ERB response element; MBS: MIZ1 binding site; CK1: casein kinase 1. Figure is adapted and modified, with permission under CCBY 4.0, from a previous publication [22].

**Figure 2 ijms-22-07761-f002:**
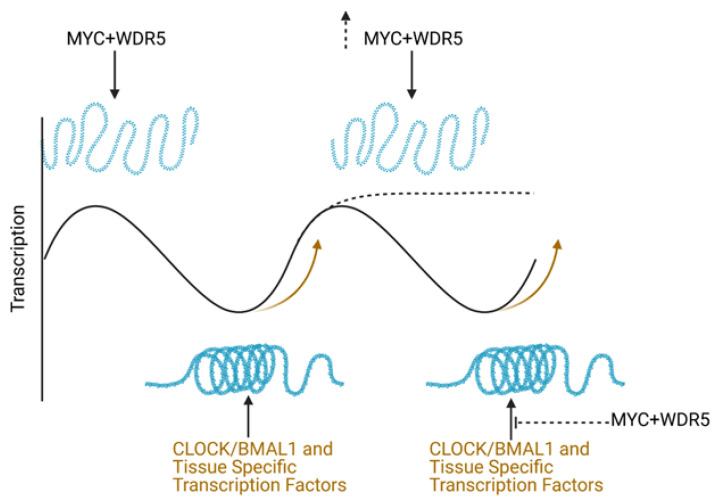
MYC and CLOCK-BMAL1 may both regulate chromatin state. Endogenous MYC facilitates transcription in a manner dependent on chromatin availability of WDR5, regulation of chromatin by CLOCK/BMAL1 complex, and other tissue specific transcription factors. This keeps chromatin open in an oscillatory fashion. When the clock is disrupted and MYC levels are increased, competition with CLOCK-BMAL1 may affect chromatin and cause the chromatin state to cease oscillation and remain more open, resulting in a higher level of transcription. Black arrows indicate MYC (in cooperation with WDR5) or CLOCK-BMAL1 contributing to chromatin state, solid black line indicates oscillatory transcription, orange arrows indicate how CLOCK-BMAL1 and tissue specific transcription factors guide this oscillation, black suppression arrow with dotted line indicates MYC (in cooperation with WDR5) interfering with CLOCK-BMAL1 control of chromatin, and dotted line indicates transcriptional oscillation (disrupted by MYC) dampening and plateauing. Created with BioRender.com, publication license acquired on 16-7-21.

**Figure 3 ijms-22-07761-f003:**
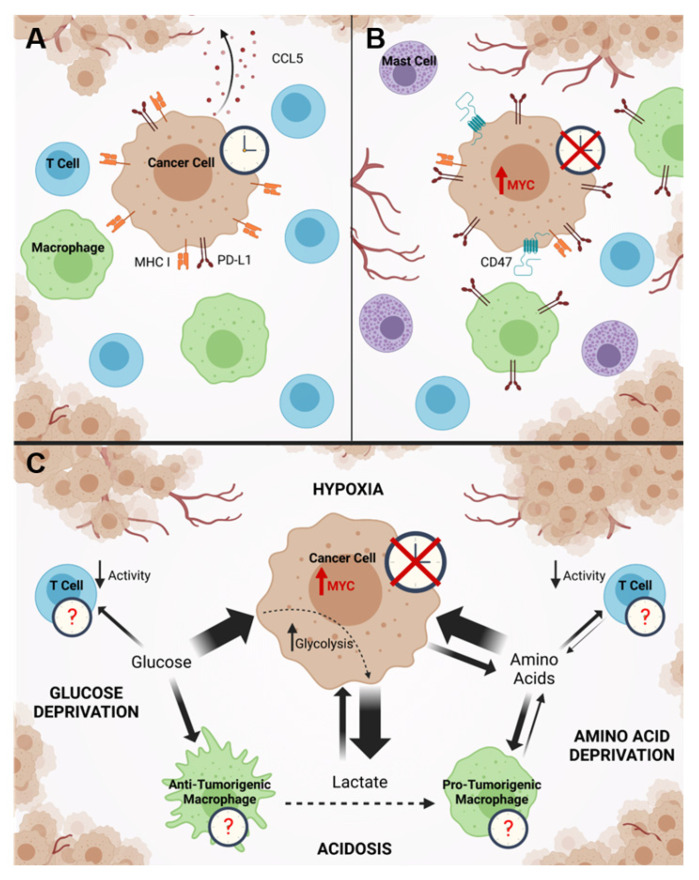
MYC drives changes in gene expression that can directly and indirectly alter the tumor immune microenvironment. (**A**). Expression of MHC I on cancer cells promotes tumor antigen-specific immune responses, while secretion of chemokines such as CCL5 mediates recruitment of immune cells to the tumor site. Low surface expression of PD-L1 within the tumor facilitates antitumor T cell activity, contributing to an immunogenic or immunologically “hot” tumor microenvironment. (**B**). MYC suppresses MHC I and upregulates PD-L1 on cancer cells, which work synergistically to suppress antitumor immune responses. MYC also suppresses CCL5 secretion, upregulates the “don’t eat me” signal CD47, and promotes recruitment of mast cells. Together, this leads to an altered composition of the tumor-immune infiltrate and skewing toward an immunosuppressive or immunologically “cold” tumor microenvironment. (**C**). MYC in cancer cells promotes cancer cell-intrinsic changes in metabolism, which can lead to changes in the availability of metabolic intermediates in the tumor microenvironment. Increased glucose and amino acid metabolism in MYC-overexpressing cancer cells drives increased uptake and subsequent depletion of these key metabolites within the immediate microenvironment. Deprived of glucose or certain amino acids, T cell activity is decreased. Simultaneously, acidosis due to increased lactate excretion polarizes tumor-associated macrophages toward a pro-tumorigenic phenotype. In conjunction with hypoxia often present within regions of solid tumors, these metabolically stressful conditions may foster an immunosuppressive microenvironment whose effects on the molecular clock of infiltrating immune cells remains unknown. In each panel, dotted line indicates polarization toward a different phenotype. Line thickness corresponds to the relative magnitude of intake or output of metabolites by cell types. Created with BioRender.com, publication license acquired on 16-7-21.

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
