# Peer review of "MYC Ran Up the Clock: The Complex Interplay between MYC and the Molecular Circadian Clock in Cancer"

_ijms, 2021, doi:10.3390/ijms22147761_

Round 1

Reviewer 1 Report

Altman and co-authors presents a review of how MYC and the molecular clock are interconnected in normal tissues and cancer. Concerning cancer, they focus on three processes (1) gene expression (2) metabolism (3) immune evasion, and finish by discussing potential clinical and therapeutic implications.

The field is particularly challenging and in constant evolution, this is a timely review that summarize the current knowledge and outline discrepancies, and, where possible, provides potential explanation to apparently contradictory data. Since many issues are still unresolved this review is in many parts highly speculative.

I would suggest the authors to revise some parts, as reported below,  in order make it easily readable and understandable by the “general” reader.

Major points

Line 172: It is not clear what the Authors mean by its, are they implying that MYC represses the circadian rhythm only if genomically rearranged and not if simply over-expressed? Please clarify.

I guess this is discussed in paragraph 2.3, if so this should be indicated in the text.

Line 233-235: please comment why deletion of Cry2 in fibroblast does not affect MYC protein level/oscillation. Is this due to differences between normal and tumor cells? If so what are these differences? See the comment above…

Chapter “3.1. Role of MYC in chromatin and global pause release ” The chapter is perhaps too long and is difficult to see a direction. Given that most of it is required to set the rationale of Paragraph 3.3 is suggest to shorten it and make it more focused.

In particular, concerning Paragraph 3.3, I think this paragraph should be re-considered given that it is highly hypothetical and there is little data supporting the model by which the interaction of MYC and the molecular clock may regulate global transcription by controlling chromatin accessibility (the molecular clock) and pause release (MYC). I get the idea, yet I suggest to the Authors to shorten it and make clear that all of this is highly speculative.

Line 286: while there is evidence for WDR5 role in regulating MYC dependent transcriptional activity, I do not think there is evidence that it affects MYC specificity  (i.e. WDR5 does not alter the set of genomic loci that are bound by MYC, it simply promotes binding of MYC to the genome).

Also the Authors are not considering all the other known interactors of MYC which contributes to its activity. Is there a particular reason why the Authors focused only on WDR5?

Line367: is this sentence supported by data? Otherwise, I think the sentence should be removed.

Chapter 6.2 is mostly reporting the current challenges in measuring circadian rhythm in clinical settings. I think the title of the paragraph should be changed in order to better reflects it content.

Minor points:

Line 32: sentence to be rephrased, not clear.

Line 157: check synthax

Line 167: substitute Per with PER

Line 234: delete “mode”

Line 293: please revise the syntax of this sentence.

Line 654: what is chromatin availability? Perhaps the Authors meant accessibility?

Line 655: sentence is convoluted and not easy to read or understand.

Author Response

We greatly thank Reviewer #1 for helpful comments and suggestions on scope and content, and have responded to all comments below.

Major points

Line 172: It is not clear what the Authors mean by its, are they implying that MYC represses the circadian rhythm only if genomically rearranged and not if simply over-expressed? Please clarify.

I guess this is discussed in paragraph 2.3, if so this should be indicated in the text.

A sentence has been added after line 172 to indicate that this topic is discussed in more detail in Section 2.3

Line 233-235: please comment why deletion of Cry2 in fibroblast does not affect MYC protein level/oscillation. Is this due to differences between normal and tumor cells? If so what are these differences? See the comment above…

In section 2.2, we have clarified that CRY2 deletion in the spleen had no effect on MYC protein levels, and that the role of CRY2 in MYC protein stability seems to be context specific.  Note that CRY2 deletion did regulate MYC protein in fibroblasts but not spleen, so it is not a matter of tumor vs non-tumor.

Chapter “3.1. Role of MYC in chromatin and global pause release ” The chapter is perhaps too long and is difficult to see a direction. Given that most of it is required to set the rationale of Paragraph 3.3 is suggest to app://resources/notifications.htmlshorten it and make it more focused.

Section 3.1 has been shortened to de-emphasize MYC’s role in remodeling chromatin and to focus on the fact that its activity relies on chromatin state.  The paragraph on pause release has also been shortened.

In particular, concerning Paragraph 3.3, I think this paragraph should be re-considered given that it is highly hypothetical and there is little data supporting the model by which the interaction of MYC and the molecular clock may regulate global transcription by controlling chromatin accessibility (the molecular clock) and pause release (MYC). I get the idea, yet I suggest to the Authors to shorten it and make clear that all of this is highly speculative.

Section 3.2 was shortened to make it more focused on CLOCK-BMAL1. Section 3.3 was somewhat shortened, and especially in the first paragraph we carefully pointed out where we were speculating.

Line 286: while there is evidence for WDR5 role in regulating MYC dependent transcriptional activity, I do not think there is evidence that it affects MYC specificity  (i.e. WDR5 does not alter the set of genomic loci that are bound by MYC, it simply promotes binding of MYC to the genome).

Thank you for pointing this out.  This sentence has been changed as follows: “Several studies have identified that WDR5 presence on chromatin is critical in permitting MYC to bind to DNA, and for MYC-driven tumorigenesis”

Also the Authors are not considering all the other known interactors of MYC which contributes to its activity. Is there a particular reason why the Authors focused only on WDR5?

We intentionally chose to focus some attention on WDR5 since it has also been shown to play a role in PER-mediated repression of CLOCK-BMAL1 (Brown et al Science 2005).  This has been added to Section 3.1.

Line367: is this sentence supported by data? Otherwise, I think the sentence should be removed.

This sentence has been deleted.

Chapter 6.2 is mostly reporting the current challenges in measuring circadian rhythm in clinical settings. I think the title of the paragraph should be changed in order to better reflects it content.

The title has been revised to better reflect that it is mostly about strategies to detect molecular clock rhythmicity in tumors and other tissues.

Minor points:

Line 32: sentence to be rephrased, not clear.

Has been rephrased to for clarity.

Line 157: check synthax

Has been rephrased for clarity.

Line 167: substitute Per with PER

Fixed

Line 234: delete “mode”

Deleted

Line 293: please revise the syntax of this sentence.

This has been revised and simplified.

Line 654: what is chromatin availability? Perhaps the Authors meant accessibility?

Thank you for pointing this out, has been changed to “accessibility.

Line 655: sentence is convoluted and not easy to read or understand.

This sentence has been simplified.

Reviewer 2 Report

The author presents a well written review which substantially covers the nuances of both Myc’s role in Cancers as well as the potential interactions with the Molecular Clock. The background was sufficiently detailed, providing good context for understanding their interest in the cross-regulation of these two pathways. This review contains a broad selection of literature which are both current and well tested.

One point that lacks resolution in this review, is the observation where Cry1/Cry2 deletion leads to Myc repression, presumably through constitutive activity of Bmal1 (line 223), while another cited study claims that Cry2 deletion results in Myc accumulation through a loss of ubiquitination and proteasomal dependent degradation (line 231).

There are many conflicting theories on the role that Circadian Clocks play in Cancers, particularly Myc-driven ones. The author suggests a compelling means of deconvolution by first classifying the cancers according to the cause of Myc abundance. We may then observe consistencies in whether maintaining Circadian Rhythms are then pro-, anti- or inconsequential to these different classes of Myc-driven cancers.

Author Response

We are very appreciative to Reviewer #2 for comments and suggestions on clarifying the role of CRY2 deletion on MYC regulation.  We responded to this comment below.

One point that lacks resolution in this review, is the observation where Cry1/Cry2 deletion leads to Myc repression, presumably through constitutive activity of Bmal1 (line 223), while another cited study claims that Cry2 deletion results in Myc accumulation through a loss of ubiquitination and proteasomal dependent degradation (line 231).

In section 2.2, we have clarified that CRY2 deletion in the spleen had no effect on MYC protein levels, and that the role of CRY2 in MYC protein stability seems to be context specific.  Note that CRY2 deletion did regulate MYC protein in fibroblasts but not spleen, so it is not a matter of tumor vs non-tumor.